# Comparison of clonal architecture between primary and immunodeficient mouse-engrafted acute myeloid leukemia cells

Naomi Kawashima [1], Yuichi Ishikawa [1], Jeong Hui Kim[1], Yoko Ushijima[1], Akimi Akashi[1], Yohei Yamaguchi[1,5], Hikaru Hattori[1], Marie Nakashima[1], Seara Ikeno[1], Rika Kihara[2], Takahiro Nishiyama[3], Takanobu Morishita[4], Koichi Watamoto[2], Yukiyasu Ozawa[4], Kunio Kitamura[3] & Hitoshi Kiyoi [1]✉

Patient-derived xenografts (PDX) are widely used as human cancer models. Previous studies demonstrated clonal discordance between PDX and primary cells. However, in acute myeloid leukemia (AML)-PDX models, the significance of the clonal dynamics occurring in PDX remains unclear. By evaluating changes in the variant allele frequencies (VAF) of somatic mutations in serial samples of paired primary AML and their PDX bone marrow cells, we identify the skewing engraftment of relapsed or refractory (R/R) AML clones in 57% of PDX models generated from multiclonal AML cells at diagnosis, even if R/R clones are minor at <5% of VAF in patients. The event-free survival rate of patients whose AML cells successfully engraft in PDX models is consistently lower than that of patients with engraftment failure. We herein demonstrate that primary AML cells including potentially chemotherapy-resistant clones dominantly engraft in AML-PDX models and they enrich pre-existing treatment-resistant subclones.

[1] Department of Hematology and Oncology, Nagoya University Graduate School of Medicine, Nagoya, Japan. [2] Department of Hematology, Komaki City Hospital, Komaki, Japan. [3] Division of Hematology, Ichinomiya Municipal Hospital, Ichinomiya, Japan. [4] Department of Hematology, Japanese Red Cross Nagoya First Hospital, Nagoya, Japan. [5]Deceased: Yohei Yamaguchi. ✉email: kiyoi@med.nagoya-u.ac.jp

Relapse and/or chemotherapy resistance are major obstacles to long-term survival in patients with acute myeloid leukemia (AML)[1–3]. Leukemia-founding clones acquire additional cooperating mutations, yielding subclones that contribute to leukemic progression and/or relapse[4,5]. Previous studies reported that acute leukemia comprises genetically diverse subclones, some of which survive after chemotherapy and contribute to disease recurrence[6,7]. Genetic analyses of samples from patients who relapsed after chemotherapies showed that a major selection pressure was imposed on leukemia clones, leading to clonal evolution[8]. A similar genetic analysis of purified subpopulations and their xenografts from paired diagnosis and relapse samples demonstrated that therapy-resistant cells were already present at diagnosis and evolved at relapse[9].

Since significant patient-to-patient cell heterogeneity complicates the clarification of a common mechanism that controls the biology of AML, a faithful model reflecting the complex heterogeneity of human AML in vivo will contribute to a more detailed understanding of the molecular pathogenesis. Many groups have developed mouse leukemia models, such as those induced by chemicals, viral infection, or irradiation, and transgenic animals expressing AML-associated proto-oncogenes[10–12]. More recently, xenograft models using immunodeficient mice enabled the ex vivo maintenance and expansion of primary leukemia cells. Patient-derived xenografts (PDX) are widely used as human cancer models mimicking the phenotypes of primary cancer cells[13–15]. However, PDX models do not always recapitulate the genetic features of primary cancer cells and polyclonal engraftment has been observed in various solid cancer PDX models[16,17]. Although multiple subclones proliferate in a patient, relapse is based on the selection of existing clones or acquisition of additional mutations during disease progression. Therefore, each genetically polyclonal population from cancer cells transplanted into PDX has a different growth ability in xenografts[6,18–20]. In a previous study on DNA copy number alteration profiling and sequencing in colorectal cancer PDX models, the proliferation, persistence, and chemotherapy tolerance of lentivirally marked lineages varied within each clone, and chemotherapy promoted the dominance of previously minor lineages in PDX[21]. Furthermore, one of the phenotypes representing the quiescent clones that expanded in serial passages of PDX was shown to be responsible for resistance to chemotherapy. A genomic analysis of 15 breast cancer PDX models revealed that polyclonal engraftment was possible; however, clonal selection in PDX was clearly evident[22]. In another study that systematically analyzed the landscapes of aneuploidy and conducted a copy number analysis of a large cohort of PDX models across multiple human cancers, the landscapes of PDX also markedly diverged from those of the parental tumors during passaging, and the genomic stability of PDX was associated with their responses to chemotherapy or targeted drugs[23]. Thus, transplanted tumor cells display a variable potential for proliferation and therapy tolerance in PDX models.

In the field of acute leukemia, a previous study demonstrated that 24 out of 48 paired primary tumor and PDX samples showed various degrees of clonal discordance based on variant allele frequency (VAF) changes[24]. However, in AML-PDX models, the significance of the clonal dynamics that occur in PDX and the role of clonal selection between parental patients and their PDX models remain unclear[25].

In the present study, by generating 160 AML-PDX models from adult AML patients and tracking the dynamics of somatic mutations in serial samples from primary AML cells and their PDX, we show their clonal architecture and the PDX-specific enrichment of AML clones with predictive power for characterizing clonal selection in parental patients.

## Results

### Clinical and genetic features of AML cells associated with engraftment into PDX.
We hypothesized that treatment-resistant clones in primary AML cells may engraft and expand in immunodeficient mice. To compare the features of primary AML cells that successfully engraft and proliferate in PDX to those with engraftment failure, we generated AML-PDX models from 160 AML patients. Mononuclear cells isolated from the fresh bone marrow (BM) or peripheral blood (PB) samples of AML patients were intravenously injected into 6-week-old NOD/Shi-scid, IL-2Rγnull (NOG) mice (purchased from the Central Institute for Experimental Animals, Tokyo, Japan) at $1–15 \times 10^6$ viable cells per mouse. The engraftment of primary samples was confirmed when the BM human CD45$^+$ percentage was higher than 20% and serial transplantation at $1 \times 10^6$ cells was successful. Patient characteristics are summarized in Table 1. PDX models from 105 patients (66%), including 84 with de novo AML, were successfully established, whereas the engraftment of human CD45$^+$ cells was not confirmed in samples from the other 55 patients (34%); 28 models died within 365 days (median 270 days, range 110–364) without positive hCD45$^+$ in PB, 18 with negative human CD45$^+$ in PB until day 365, 7 with T-cell engraftment, and 2 failed in serial passaging with hCD45$^+$ >5% in PB, but <20% in BM, at day 365. Engraftment was achieved within a median of 106 days (range, 27–380 days) post-transplantation, and was more frequently observed in mice transplanted with more than $5.0 \times 10^6$ cells (71% vs. 38%, $P = 0.001$). Successful engraftment was also associated with the M4-FAB type ($P = 0.028$), relapse/refractory (R/R) AML ($P < 0.001$), and a higher risk in the European LeukemiaNet (ELN) classification (adverse, 74.7%; intermediate, 55.2%; favorable, 41.7%; $P = 0.022$), but not with patient age ($P = 0.621$), cytogenetic risks ($P = 0.053$), graft sources ($P = 0.060$), or infused blast percentages ($P = 0.280$).

### Genetic landscape of engrafted primary AML cells.
To obtain further insights into the genetic landscapes of engrafted primary AML cells, the mutation profiles of AML cells were analyzed. Among cells from 160 AML patients, the targeted sequencing of 54 genes (Supplementary Table 1) was performed using bulk PB or BM samples from 69 AML patients that successfully engrafted into NOG mice and 46 AML patients with engraftment failure. The other 36 AML samples with engraftment and 9 samples with engraftment failure were excluded from subsequent genetic analyses because the quality of patient DNA was insufficient or they were derived from different time points of the same patients. Among the evaluated genes that are frequently mutated in myeloid malignancies, mutations in the *FLT3* (59% vs. 15%, $P < 0.0001$), *NPM1* (38% vs. 4%, $P < 0.0001$), *IDH1* (15% vs. 0%, $P = 0.007$), and *WT1* genes (13% vs. 2%, $P = 0.04$) were more frequently observed in engrafted patients than in those with engraftment failure (Fig. 1). Mutations in the *RAD21* (16% vs. 4%, $P = 0.05$), *ASXL1* (15% vs. 7%, $P = 0.19$), *RUNX1* (15% vs. 7%, $P = 0.19$), and *DNMT3A* (25% vs. 15%, $P = 0.22$) genes were also frequent, but not significantly different. In contrast, AML cells with *RUNX1-RUNX1T1* (0% vs. 7%, $P = 0.03$) and *CBFB-MYH11* (1% vs. 9%, $P = 0.06$) showed lower engraftment rates. These results revealed the existence of specific clones that have the ability to engraft and proliferate in AML-PDX models as well as their genetic profiles.

### PDX-specific clonal engraftment from primary AML cells.
We then investigated whether AML-PDX recapitulated the molecular phenotypes of primary leukemia. To assess changes in phenotypes, we analyzed the cell surface markers of leukemic stem/progenitor cells using a flow cytometric analysis of CD34- and CD38-

**Table 1 Characteristics of patients' samples.**

| Variables | | Engraftment | Failure | P-value[a] |
|---|---|---|---|---|
| Number (%) | | 105 (66%) | 55 (34%) | |
| Patient age, median (range) | | 65 (18–92) | 64 (24–86) | 0.621 |
| Disease | de novo AML | 84 | 41 | 0.373 |
| | FAB classification | | | |
| | M0 | 2 | 1 | 0.028 |
| | M1 | 15 | 5 | |
| | M2 | 14 | 17 | |
| | M3 | 8 | 5 | |
| | M4 | 38 | 9 | |
| | M5 | 6 | 3 | |
| | M6 | 1 | 0 | |
| | M7 | 0 | 1 | |
| | AML with MRC | 12 | 12 | |
| | tAML from MPN | 6 | 2 | |
| | AL of ambiguous lineage | 2 | 0 | |
| | CML-BC | 1 | 0 | |
| Cytogenetic risk | Favorable | 10 | 14 | 0.053 |
| | Intermediate | 48 | 25 | |
| | Poor | 37 | 16 | |
| | Not examined | 10 | 0 | |
| ELN risk category[b] | Favorable | 10 | 11 | 0.022 |
| | Intermediate | 21 | 17 | |
| | Adverse | 59 | 20 | |
| | unknown | 7 | 2 | |
| Graft source | BM/PB | 69/36 | 44/11 | 0.060 |
| Sample | New/relapse/refractory | 58/29/18 | 48/5/2 | <0.001 |
| Infused cell count | $\geq 5 \times 10^6$ | 95 | 39 | 0.001 |
| | $< 5 \times 10^6$ | 10 | 16 | |
| Infused blast %, median (range) | | 58.4 (1–100) | 46.3 (3–99.5) | 0.280 |
| Engraft day, median (range) | | 106 (27–380) | - | - |

[a]Chi-square test for Graft source, Fisher's exact test for other categorical data and the Mann–Whitney U-test for continuous variables.
[b]APL excluded.

expression in 29 paired bulk primary AML cells and the human CD45[+] fraction of their engrafted AML-PDX BM cells (Supplementary Table 2). Nine AML-PDX models (31%) showed concordant patterns of CD34 and CD38 positivity to the parental patients (Fig. 2a). On the other hand, the hierarchy of leukemic stem/progenitor cells was discordant in the other 20 cases (69%). Specific fractions of CD34 and CD38 positivity from primary AML cells dominantly engrafted and proliferated in their PDX models (Fig. 2b, c and Supplementary Table 2). To identify the dominantly proliferative clones in AML-PDX models characterized by somatic mutations, we assessed clonal diversity by comparing the VAF of 54 genes in 69 paired primary AML cells and their engrafted PDX BM cells. Overall, 231 mutations (111 single nucleotide variants, 78 frameshifts, and 42 insertions/deletions) were identified (Supplementary Table 3). The VAF of the detected genetic variants of patient primary AML cells were not completely consistent with those of engrafted PDX BM cells ($r^2 = 0.2619$, $P < 0.0001$), supporting the existence of clonal selection in PDX models (Fig. 2d). When the VAF of each gene was compared, genes with mutations that were enriched or diminished in PDX models were identified (Fig. 2e). Variants in *IDH1* ($r^2 = 0.751$, $P = 0.0012$), *IDH2* ($r^2 = 0.986$, $P < 0.0001$), *EZH2* ($r^2 = 0.918$, $P = 0.003$), *TP53* ($r^2 = 0.774$, $P = 0.0018$), *NPM1* ($r^2 = 0.106$, $P = 0.11$), and *RUNX1* ($r^2 = 0.196$, $P = 0.13$) showed uniformity between primary AML and PDX BM cells (Fig. 2e, f and Supplementary Fig. 1a). In contrast, VAF was unstable between primary AML and PDX cells in the following gene mutations: *FLT3*-ITD ($r^2 = 0.149$, $P = 0.017$), *FLT3*-TKD ($r^2 = 0.0824$, $P = 0.42$), *WT1* ($r^2 = 0.039$, $P = 0.48$) *CEBPA* ($r^2 = 0.0377$, $P = 0.75$), *NRAS* ($r^2 = 0.001$, $P = 0.91$), *KRAS* ($r^2 = 0.0249$, $P = 0.77$), and *DNMT3A* ($r^2 = 0.3571$, $P = 0.009$). They were enriched more in

PDX than in primary AML cells (Fig. 2e, g and Supplementary Fig. 1b). On the other hand, the VAF of *ASXL1* ($r^2 = 0.0619$, $P = 0.49$) and *PTPN11* ($r^2 = 0.094$, $P = 0.39$) were slightly higher in primary AML cells and decreased when engrafted in PDX models (Fig. 2e, g and Supplementary Fig. 1b). These results confirmed the existence of PDX-specific clonal selection in AML-PDX. In 11 patients with chimeric transcripts (Fig. 1), they were preserved in all engrafted cells.

**Relapsed AML exhibits few clonal changes in their PDX models.** To assess the degree of clonal dynamics from primary AML to engrafted PDX BM cells according to the disease status of transplanted primary AML cells, we compared changes in the VAF of 50 pairs of newly diagnosed AML patients and their PDX BM cells and 31 pairs of R/R AML patients and PDX samples (Fig. 3a, b). Although changes in VAF were observed in many PDX models of both newly diagnosed and R/R samples, a 10-fold or higher increase in VAF was more frequently observed in PDX models of newly diagnosed samples (22/169 variants) than in those of R/R samples (8/137 variants, $P = 0.033$, Fig. 3c). These results suggest that newly diagnosed AML cells contained more minor clones with growth potential in NOG mice than R/R samples.

Since minor leukemic subclones may require time to selectively expand in PDX models, we evaluated the time from transplantation to engraftment according to the patient disease status and mutational landscapes of transplanted cells. No significant differences were observed in the speed of engraftment among the mutated gene families (Supplementary Fig. 2a). Comparisons of every variant detected in PDX models revealed that those carrying mutations in the *EZH2* and *KRAS* genes showed

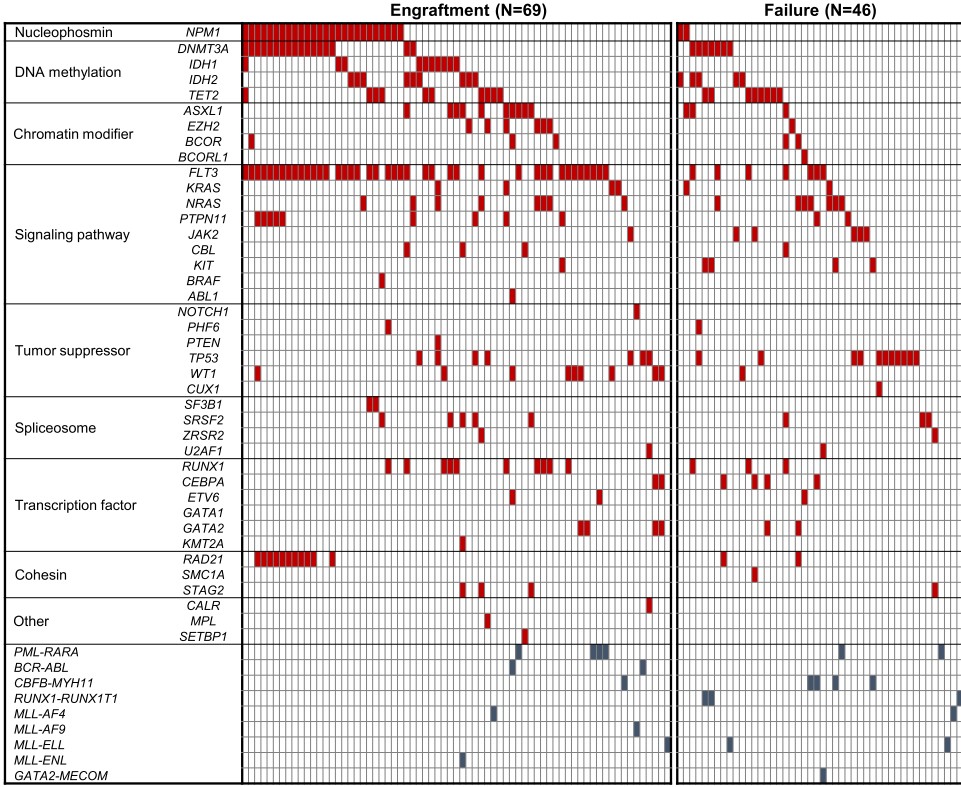

**Fig. 1 Mutational landscape of 115 primary AML patients according to the engraftment status in NOG mice.** Each column represents each patient. Signaling, epigenetic, and cohesin genes are separated into functional groups.

significantly shorter durations until engraftment, the medians of which were 62 days (range 58–83, $P = 0.010$) and 72 days (median 29–147, $P = 0.036$), respectively (Supplementary Fig. 2b). In comparisons among newly diagnosed ($N = 58$), relapse ($N = 29$), and refractory ($N = 18$) AML, relapsed AML (median 102 days, range 45–380 days, $P = 0.049$) and refractory AML (median 76 days, range 27–159 days, $P = 0.0002$) provided faster engraftment than newly diagnosed AML (median 138 days, range 29–362 days), which is consistent with few clonal changes in relapsed AML-PDX models. Refractory AML showed the fastest engraftment, suggesting only slight clonal selectivity in the status of these patients (Fig. 3d). Engraftment latency widely ranged between 27 and 380 days. The time to engraftment was associated with the ELN risk categories (adverse risk group; median 104 days, range 45–362 days, intermediate risk group; median 126 days, range 27–196 days, favorable risk group; median 161 days, range 106–330 days, $P = 0.036$), but not with FAB classifications, cytogenetic abnormalities, or the infused blast burden (Supplementary Table 4).

**Clonal selection through propagation in AML-PDX.** The present results prompted us to speculate that the clones dominantly proliferating in PDX models may be further selected and isolated through passages of mice. To obtain further insights into clonal selection in AML-PDX models through propagation, changes in VAF were sequentially compared among primary AML cells, first engrafted or once passaged (P1-2) PDX models, and further passaged (P3-11) PDX models (Fig. 4a). Engrafted patterns were categorized into three subgroups according to clonal sizes: the engrafted clone was enriched (Fig. 4b), the same clonal size as primary AML cells (Fig. 4c), or diminished (Fig. 4d) in PDX models. Except for variants with stable VAF between primary AML and engrafted PDX BM cells, those expanded in P1 PDX were further enriched during serial PDX passages. The variants

that decreased in P1 PDX also diminished through passages. Enriched genetic variants frequently included the *FLT3*, *TP53*, *KRAS*, *TET2*, and *WT1* mutations (Fig. 4b). In contrast, clones harboring *NRAS* or *CEBPA* occupied small VAF at engraftment and then diminished through serial passages in PDX models (Fig. 4d).

To assess individual differences between NOG mice, we transplanted primary AML cells with the same cell count into duplicated mice and evaluated discrepancies in engrafted clones using the VAF of each variant in five AML-PDX models (Supplementary Fig. 3a). VAF changes in each variant in duplicated AML-PDX models are shown in Supplementary Fig. 3b. Although gains or losses in VAF were observed between patients and their PDX cells, as shown in Fig. 2d, we detected comprehensive concordance in the VAF of each gene between these duplicated PDX models, which was maintained even after propagation (PDX#360 and #789 in Supplementary Fig. 3b).

**Engraftment potency in PDX represents poor responses to chemotherapy in primary AML patients.** We attempted to establish whether clones that engraft and expand in PDX actually exert adverse effects on the clinical outcomes of patients, such as treatment resistance. We also evaluated the responses of patients to chemotherapy according to the potential to engraft in NOG mice. Event-free survival (EFS) was analyzed in 76 patients who received cytotoxic chemotherapies and whose primary AML cells at diagnosis were transplanted into PDX (Fig. 5). The median follow-up period of survivors was 376 days (range, 32–1754 days). Thirty-nine patients with successful engraftment into PDX had a significantly lower EFS rate than 37 patients with engraftment failure ($P = 0.0004$), with 2-year EFS rates of 20.0% (95% confidence interval [CI], 7.2–37.3%) and 53.5% (95%CI, 33.2–70.1%), respectively. Two-year overall survival (OS) and cumulative incidence of relapse rates were also significantly poorer in PDX-

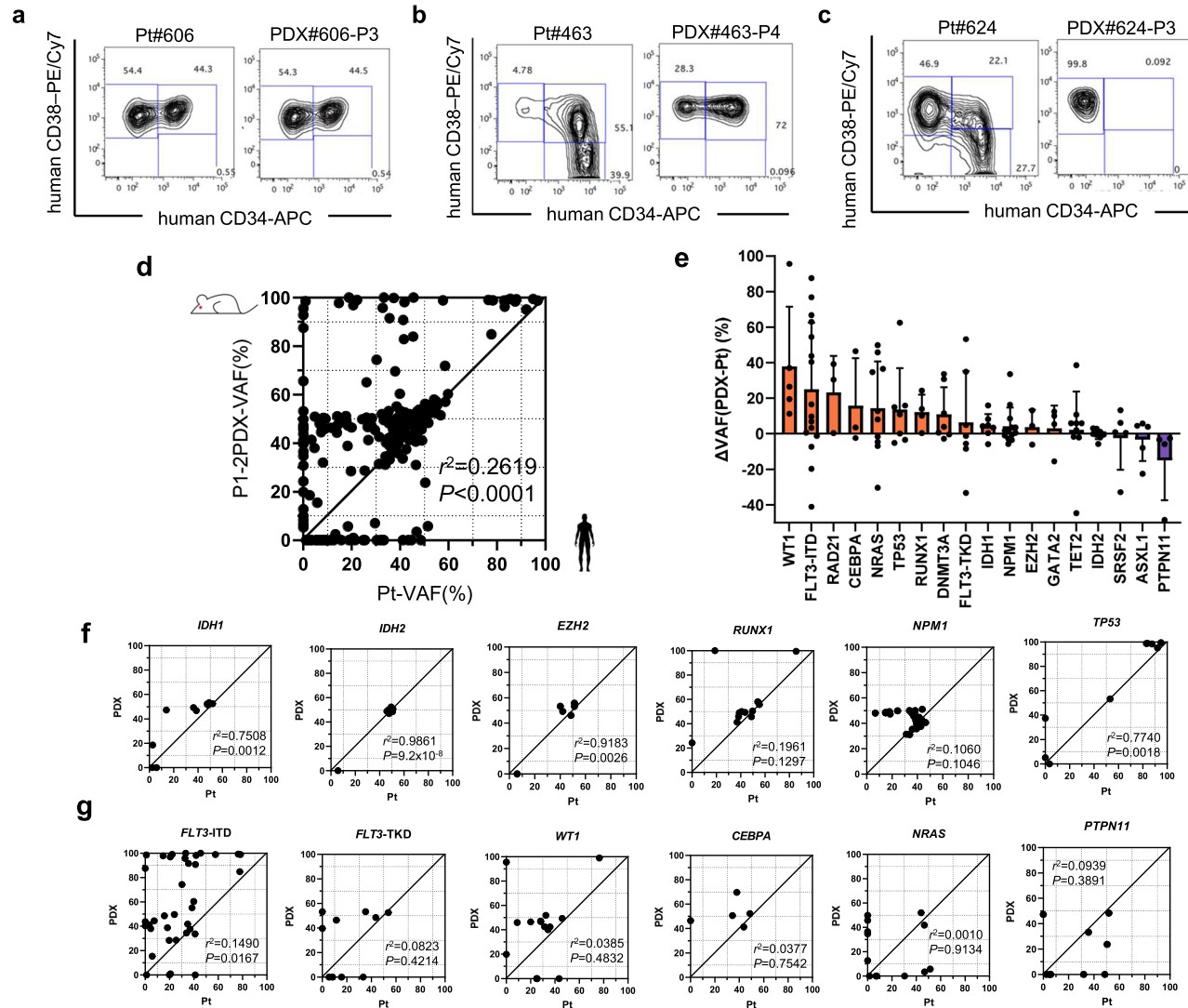

**Fig. 2 Diversity in molecular phenotypes between primary AML and their corresponding PDX models. a–c** Flow cytometric analysis of hematopoietic stem cells by the expression of CD34 and CD38 in representative pairs of primary AML patients and AML-PDX models. **d** Scatter plots showing the variant allele frequencies (VAF) relationship between primary AML cells and P1-2 PDX BM cells. Every dot represents each single genetic alteration. $R^2$ value and two-tailed $P$-value calculated using Pearson's correlation indicate the strength and significance, respectively, of the relationships. **e** Waterfall plots show changes in VAF between primary AML and PDX. Genes in which mutations were detected in more than three patients are shown (*WT1*, $n = 5$; *FLT3*-ITD, $n = 16$; *RAD21*, $n = 3$; *CEBPA*, $n = 3$; *NRAS*, $n = 10$; *TP53*, $n = 7$; *RUNX1*, $n = 4$; *DNMT3A*, $n = 7$; *FLT3*-TKD, $n = 7$; *IDH1*, $n = 7$; *NPM1*, $n = 13$; *EZH2*, $n = 3$; *GATA2*, $n = 4$; *TET2*, $n = 9$; *IDH2*, $n = 8$; *SRSF2*, $n = 5$; *ASXL1*, $n = 5$; *PTPN11*, $n = 4$ biologically independent samples). Bars, median; Error bars, standard deviations. **f** Scatter plots of VAF relationships between primary AML cells and P1-2 PDX BM cells in genes with strong correlations. **g** Scatter plots of VAF relationships between primary AML cells and P1-2 PDX BM cells in genes with weak correlations. $R^2$ values and two-tailed $P$-values calculated using Pearson's correlation indicate the strength and significance, respectively, of the relationships. Source data are provided as a Source Data file.

engrafted patients, at 28.8% (95%CI, 10.5–50.3%) vs. 50.9% (95% CI, 22.3–73.9%, $P = 0.045$) and 74.6% (95%CI, 54.8–86.7%) vs. 45.4% (95%CI, 27.6–61.6%, $P = 0.029$), respectively (Supplementary Fig. 4). These results indicate that primary AML cells including potentially chemotherapy-resistant clones engraft in AML-PDX models.

Based on these results, we hypothesized that treatment-resistant clones may be identified by investigating engrafted clones in AML-PDX models generated from patients at diagnosis.

**Enriched AML subclones in PDX models recapitulate clonal selection in patients acquiring treatment resistance.** To elucidate whether selectively expanded subclones in PDX models reflect clonal evolution in the corresponding primary AML

patients, we evaluated serial clonal dynamics in 21 newly diagnosed AML patients by analyzing patient samples at diagnosis and R/R and each corresponding PDX model (Supplementary Table 5). Dominant clones were identified from the VAF of each variant in each sample. Of note, selectively expanded subclones in PDX generated from AML samples at diagnosis (Diagnosis-PDX) were predictively consistent with the dominant clones at R/R in 12 out of 21 (57%) patients (Fig. 6 and Supplementary Fig. 5).

Patient #28 had subclones carrying $RUNX1^{R107HWR}$ (VAF, 38.8%) and $WT1^{C453Y}$ (VAF, 20.0%) and/or $IDH1^{R132H}$ (VAF, 5.9%) at the diagnosis of AML-M2 (Fig. 6a). The number of clones carrying the $IDH1^{R132H}$ mutation was smaller in the Diagnosis-PDX model than in this patient's sample, whereas clones carrying the 15-bp insertion of the $FLT3$-ITD mutation, which was not detected in the patient's BM sample at diagnosis,

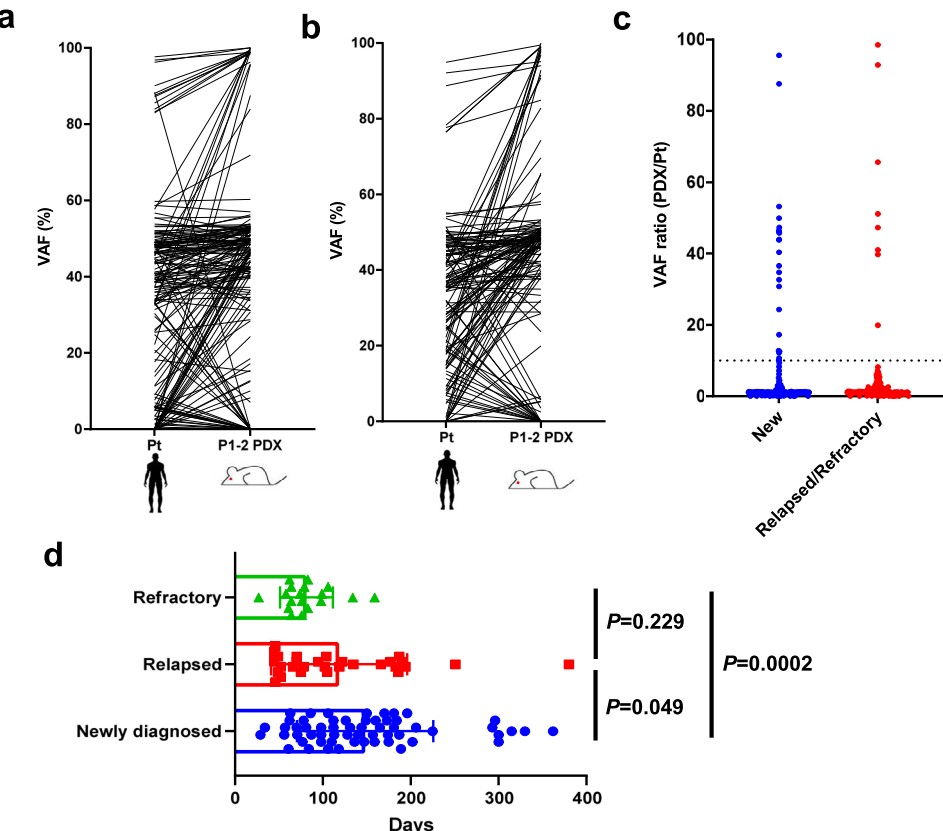

**Fig. 3 Stable and rapid engraftment of relapsed/refractory AML into PDX.** Changes in VAF from primary to PDX in samples from **a** newly diagnosed and **b** relapsed/refractory AML. **c** The ratio of VAF changes from patients to PDX (New, $n = 169$; Relapsed/Refractory, $n = 137$). The dotted line shows a cut-off of 10. **d** Duration from transplantation to engraftment in PDX of primary patients at diagnosis ($n = 58$), relapse ($n = 29$), or refractory ($n = 18$) to initial chemotherapy. Data are presented as mean values $+/-$ standard deviations. $P$-values indicate significance from the Mann–Whitney $U$-test. Source data are provided as a Source Data file.

emerged to VAF of 43.2% accompanied by $RUNX1^{R107HWR}$ (VAF, 49.0%) and $WT1^{C453Y}$ (VAF, 46.4%). This clonal architecture was consistent with that of the patient at relapse after consolidation therapy with $RUNX1^{R107HWR}$ (VAF, 48.9%), $WT1^{C453Y}$ (VAF, 45.8%), and $FLT3$-ITD (VAF, 77.7%). On the other hand, PDX generated from this relapsed patient's sample (Relapse-PDX) did not show further clonal selection, but maintained similar clonality to patient BM samples with $RUNX1^{R107HWR}$ (VAF, 45.7%), $WT1^{C453Y}$ (VAF, 49.4%), and $FLT3$-ITD (VAF, 84.9%). Patient #266 showed the same pattern of clonal diversity (Fig. 6b). BM AML cells at diagnosis consisted of heterogeneous subclones harboring mutations in $CEBPA$ (E309dupGAG 43.7% and S28Afs 48.8%), $GATA2^{R330L}$ (VAF, 47.7%), and $WT1^{L378Dfs}$ (VAF, 8.9%). In the Diagnosis-PDX model, the clone carrying all of these mutations expanded ($CEBPA^{E309dupGAG}$ 41.3%, $CEBPA^{S28Afs}$ 52.3%, $GATA2^{R330L}$ 53.1%, and $WT1^{L378Dfs}$ 46.0%). Once hematological remission was achieved post-induction chemotherapy, this patient relapsed with similar clonality to Diagnosis-PDX ($CEBPA^{E309dupGAGfs}$ 34.3%, $CEBPA^{S28Afs}$ 38.0%, $GATA2^{R330L}$ 36.0%, and $WT1^{L378Dfs}$ 35.9%).

In patient #624 (Fig. 6c), clones harboring $NPM1^{W288Cfs}$ and the 57-bp insertion of $FLT3$-ITD mutations showed similar VAF between the patient at diagnosis (37.2% and 25.4%, respectively) and Diagnosis-PDX (45.7% and 28.7%, respectively). When the patient relapsed after consolidation chemotherapy, major clones carrying $NPM1^{W288Cfs}$ and the 57-bp insertion of $FLT3$-ITD mutations persisted with similar VAF (35.4% and 39.9%, respectively).

Cases of discordance were noted between Diagnosis-PDX models and paired patient samples at relapse, with the emergence of new clones observed only in Diagnosis-PDX. In patient #606 shown in Fig. 6d, primary AML cells at diagnosis carried a single nucleotide variant in $GATA2^{A372T}$ (VAF, 46.5%) and a 48-bp insertion of $FLT3$-ITD (20.7%). When engrafted in Diagnosis-PDX, clonality markedly changed with the emergence of a frameshift mutation in $WT1^{A382Rfs}$ (VAF, 95.6%) and a single nucleotide variant in $KRAS^{G12D}$ (VAF, 43.9%) with the regression of clones carrying $GATA2^{A372T}$ (VAF, 31.0%) and $FLT3$-ITD (VAF, 1.0%). On the other hand, the relapsed AML cells of the same patient after consolidation chemotherapy harbored clones including $GATA2^{A372T}$ (VAF, 26.1%), $WT1^{D464G}$ (VAF, 76.4%), and the 48-bp insertion of $FLT3$-ITD (VAF, 19.7%). Relapse-PDX showed the concordant clonality of $GATA2^{A372T}$ (VAF, 65.1%), $WT1^{D464G}$ (VAF, 99.0%), and $FLT3$-ITD (VAF, 28.5%). In this case, although Diagnosis-PDX did not predict the clonal architecture of AML cells at relapse post-consolidation therapy, results from its relapsed sample indicated that AML cells at relapse were less heterogeneous than those before treatment, consistent with the results shown in Fig. 3b, c.

We found that minor treatment-resistant clones with lower VAF (<5%) expanded in PDX. In patient #463 (Fig. 6e), who received quizartinib as salvage treatment for refractory AML with the $FLT3$-ITD mutation, clones harboring $EZH2^{D657A}$ (VAF, 51.3%), $NRAS^{G13D}$ (VAF, 30.4%), and $FLT3$-ITD (VAF, 57.7%) were detected when quizartinib was initially administered. During the third course of administration, the patient did not respond to quizartinib and AML cells proliferated. BM samples after the first

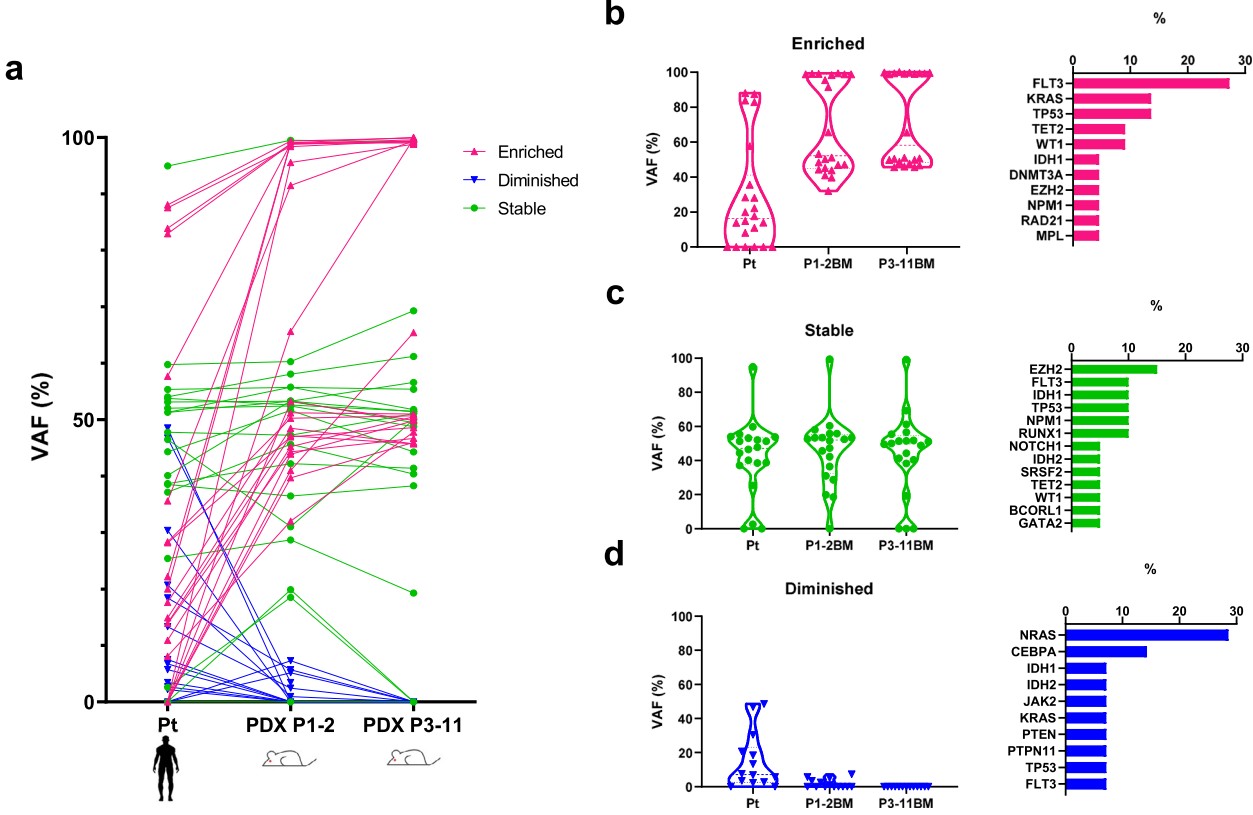

**Fig. 4 Clonal selection occurs in AML-PDX models through PDX passaging. a** VAF (%) of primary samples and PDX after serial passaging. Each dot plot indicates the same single alteration ($n = 56$). **b**–**d** Violin plots of the VAF of variants showing **b** enrichment, **c** overall similarity, and **d** reductions from primary through P1-2 to ≥P3 PDX datasets. The relative proportions of genes included in each group are shown in the bar plot. Source data are provided as a Source Data file.

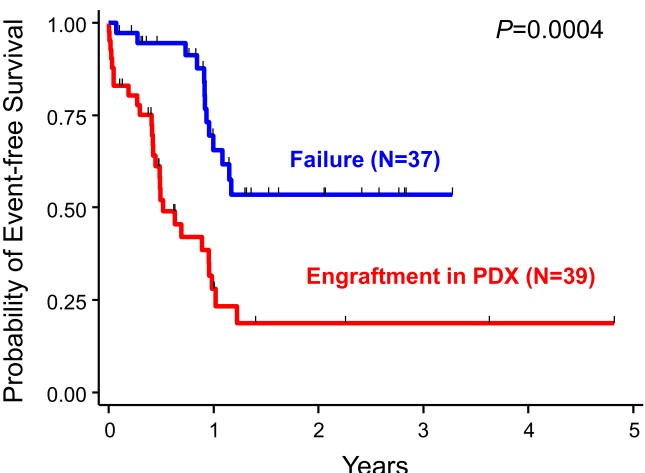

**Fig. 5 Event-free survival of AML patients according to the establishment of AML-PDX.** Kaplan–Meier estimates of event-free survival in 76 newly diagnosed AML patients according to the engraftment status in NOG mice. *P*-values were estimated from the Log-rank test. Source data are provided as a Source Data file.

course (Refractory#1) and third course of consolidation therapy (Refractory#2) showed similar clonality with the VAF of $EZH2^{D657A}$ (48.6 and 51.3%), $NRAS^{G13D}$ (51.4 and 46.8%), and $FLT3$-ITD (41.3 and 35.6%) without newly acquired mutations. However, in the PDX of the BM sample at this time of resistance (Refractory#2-PDX), the clone carrying $FLT3^{D835Y}$, a resistance

mutation to quizartinib, newly expanded with VAF of 39.7% in addition to pre-existing $EZH2^{D657A}$ (53.3%), $NRAS^{G13D}$ (3.5%), and $FLT3$-ITD (91.5%). In these cases, AML subclones selectively enriched in PDX models recapitulated clonal selection in patients acquiring treatment resistance.

## Discussion

In the present study, we generated 105 serially transplantable AML-PDX models, including a series of PDX models from sequential AML patient samples from diagnosis through to the acquisition of resistance to individual treatments. Tumor evolution and copy number alterations during PDX passaging were reported in a previous study on more than 1000 PDX models across 24 solid cancer types[23]. We consistently found that particular AML clones were selected from heterogeneous primary AML cells and enriched though the serial passaging of their AML-PDX models. These results indicate that AML-PDX models did not completely recapitulate the clonality of patients, and indicate the need for caution regarding the application of PDX.

We also demonstrated the instability of clonal sizes according to genetic variants in engrafted PDX (Fig. 2d–g and Supplementary Fig. 1). The sizes of clones harboring $PTPN11$ or $ASXL1$ mutations decreased when engrafted in PDX, and those of $CEBPA$ and $NRAS$ mutations gradually diminished through the passaging of PDX (Fig. 4d). On the other hand, clones carrying $FLT3$ or $WT1$ mutations expanded in PDX. In contrast to mutant $NRAS$ clones, mutant $KRAS$ clones expanded during the serial propagation of AML-PDX (Fig. 4b, d). Since all mutant and expanded $KRAS$ clones were considered to be co-mutated with

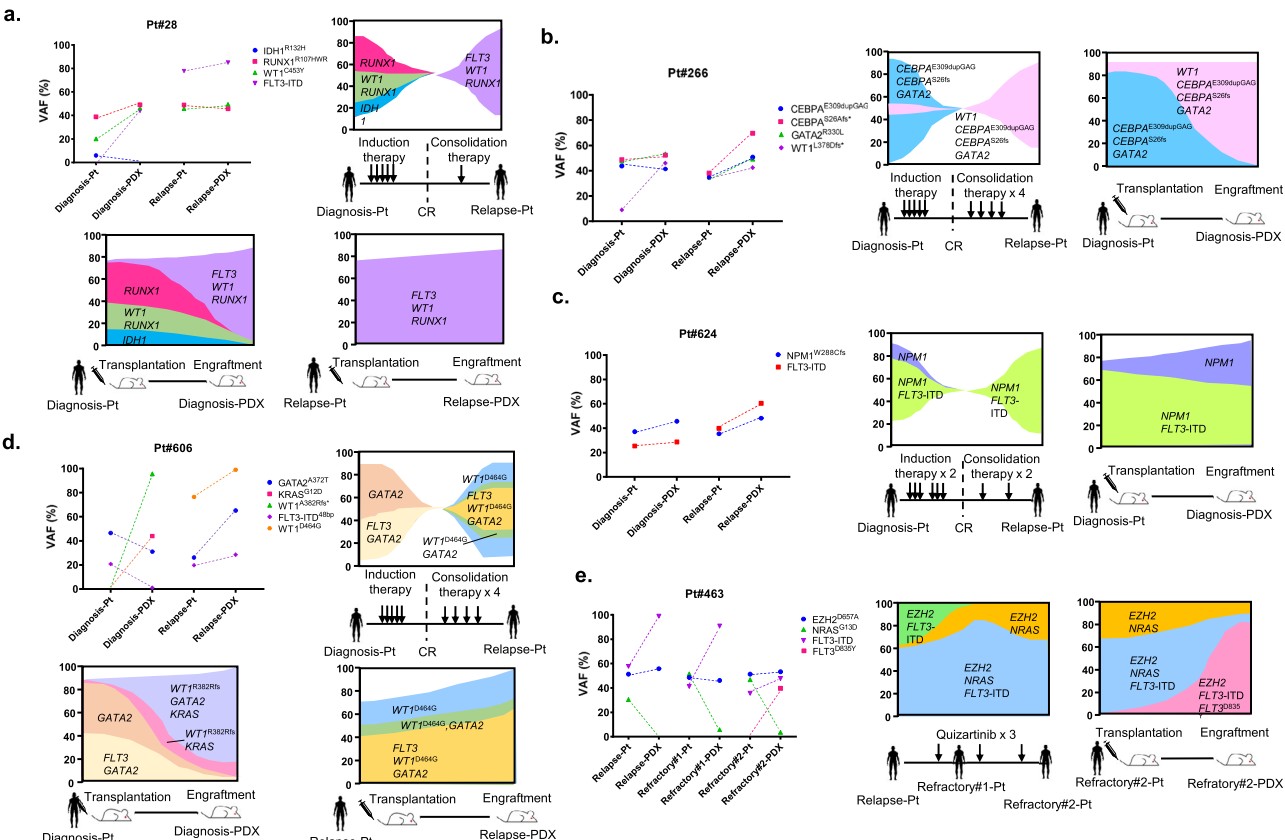

**Fig. 6 Serial mutational spectrum and clonal changes in AML patients and their PDX models.** In each patient, clonal changes in the clinical course and PDX engraftment are shown in the fish plot format. **a, b** Illustrative cases showing that selectively expanded clones in PDX were consistent with clonality at relapse in patients. **c** A case in which dominant clones consisting of *FLT3*-ITD and *NPM1* mutations persisted in patients at relapse and PDX at engraftment. **d** A case in which mouse-specific enrichment that did not evolve in a relapsed patient. **e** A case showing the expansion of a FLT3 inhibitor-resistant clone in PDX. Source data are provided as a Source Data file.

*WT1* according to their VAF, these clones may expand through propagation. A *FLT3*-ITD, *TET2*, *TP53*, or *WT1* mutation was also identified in mutant *NRAS* primary samples; however, the VAF of *NRAS* differed from those of the other genes, indicating that none of the mutant *NRAS* clones harbored a *FLT3*-ITD, *TET2*, *TP53*, or *WT1* mutation. In 16 *NRAS*-engrafted P1-2 PDX models, 11 showed the stable or enriched engraftment of mutant *NRAS* clones, whereas these clones were markedly diminished in the other 5 models because the clones carrying *FLT3*-ITD ($N = 4$), *TET2* ($N = 2$), *TP53* ($N = 1$) or *WT1* ($N = 1$) dominantly expanded.

A previous study reported that the co-mutation of the *NPM1* and *DNMT3A* genes was associated with a higher engraftment ability, whereas a single mutation was not[26]. The co-mutation of the *NPM1* and *DNMT3A* genes consistently showed a higher engraftment ability (22% vs. 0%, $P = 0.001$). In addition, most of the genes also carried *FLT3* mutations (Fig. 1). However, a single mutation in the *FLT3*, *NPM1*, *IDH1*, and *WT1* genes influenced engraftment in our PDX models. These results indicate that further mechanisms influence engraftment and selection in AML-PDX models.

The frequency of leukemic long-term culture-initiating cells, but not the extracellular phenotype, has been associated with engraftment in NSG mice[27]. Most CD34⁺ fractions and few CD34⁻ fractions were previously reported to contain leukemia stem cells (LSC)[28]. In the present study, the CD34⁻/38⁺ fraction dominantly expanded in most PDX models, while the CD34⁺/CD38⁻ or CD34⁺/38⁺ fraction decreased in engrafted PDX (Supplementary Table 2). In addition, an adverse risk in the ELN

classification was associated with shorter engraftment latency as previously reported (Supplementary Table 4)[26,29]. These results collectively indicate that cell fractions that are more differentiated than LSC also have advantages for proliferation in PDX models if they acquire genetic alterations associated with clinical prognosis.

Clonal evolution in AML is a continuous process that is expected to be markedly altered by the selective pressures applied during chemotherapy. Difficulties are associated with capturing minor clones related to treatment resistance at the time of diagnosis until they proliferate in patients. For example, a similar genetic analysis between baseline and progression samples from patients treated with gilteritinib revealed that *FLT3* mutation-negative clones newly acquired RAS/MAPK pathway mutations and expanded as a resistant clone in 12% of patients[30]. In the present study, since patients containing PDX-specifically expanded clones showed resistance to chemotherapy (Fig. 5) and selectively enriched clones in AML-PDX models notably mimicked clonal evolution in some patients at relapse after chemotherapy (Fig. 6), the present results indicate that AML-PDX models recapitulate, at least partially, the complex clonal dynamics of primary AML. In an analysis using a flow cytometry-based method for the prediction of engraftment Griessinger et al. demonstrated that survival was significantly shorter in patients who were predicted to engraft in NSG than in those not predicted to engraft, reflecting the aggressiveness of AML[27]. Culen et al.[26] reported significantly shorter OS and a trend towards shorter EFS in patients with a higher percentage of engrafted human CD45⁺ cells. More recently, Sandén et al.[31] revealed clonal dynamics in 84 PDX models using NSG mice and whole exome sequencing,

which was consistent with clonal evolution in our sequential PDX models from diagnosis to relapse.

Our AML-PDX system enriched minor clones in patients to detectable levels in PDX before they evolved in patients at relapse. This may be applied to predictions of the potential of treatment resistance in particular treatments before their initiation. As shown in Fig. 6e, subclones harboring the $FLT3^{D835Y}$ mutation, which are reportedly resistant to quizartinib[32], were expected to already exist, but with very low VAF in primary patient BM and, thus, were not detected; however, they dominantly expanded in PDX, reflecting the acquisition of quizartinib resistance. Therefore, we propose that AML-PDX models may be used not only for in vivo drug screening, but also for identifying treatment-resistant clones and elucidating the biology of AML.

On the other hand, there are limitations in the present study. Since the analysis was based on bulk AML samples, clonal sizes were assumed by the VAF of analyzed genes. Therefore, a single cell analysis is required to identify detailed clonal dynamics characterized by genetic mutations. Furthermore, in the setting of AML-PDX models using immunodeficient mice, although PDX retain the architecture found in human tumors, the tumor microenvironment, which plays an essential role in tumor progression, is not completely recapitulated[33–35]. PDX may lack the factors suppressing the proliferation of AML cells in patients. For example, tumor extrinsic factors, including reactive ligands as well as stromal and immune interactions, may be altered by interspecies compatibility and cellular component deficiencies in host models. AML clones carrying genetic variants in signaling pathways expanded well in our PDX models; however, further studies are needed to clarify whether constitutive reactivation by these ligands suppresses the proliferation of AML similarly to the natural environment in patients. Since NOG mice with macrophage inactivity and the loss of T, B, and NK cells were used in our models, cross-talk between tumor progression and immune surveillance was not mimicked. Therefore, discordance and heterogeneity between primary AML at diagnosis and their PDX may have occurred, and the subclones vulnerable to immune reactions in patients may have survived in PDX with a weaker capacity to evade immune surveillance by NK cells[36]. The eliminated clones in patients were still detected or expanded in PDX models. For example, based on our results, clones harboring mutations in the WT1 gene were one of the enriched genetic features in PDX (Fig. 2e, g); however, as reported for tumor antigens[37], there may be discordance due to humans having the capacity to eliminate WT1 mutations, whereas PDX do not. Accordingly, care is needed when interpreting whether expanded clones in PDX predict clonal selection in parental patients based on differences between relapse after chemotherapy and allogeneic hematopoietic stem cell transplantation because immunologically eliminated clones by allogeneic transplantation were not mimicked in this model.

In conclusion, we herein elucidated the dynamics of clonal changes from primary patients to AML-PDX models. Primary AML cells including potentially chemotherapy-resistant clones engraft in AML-PDX models. The skewing of minor pre-existing treatment-resistant subclones in AML-PDX, which were under the detection limit in patients, predict clonal evolution at treatment resistance. The present results will contribute to the application of AML-PDX models not only for drug discovery, but also a more detailed understanding of clonal selection in AML.

## Methods

**Patient samples and transplantation**. Patient samples were acquired from Nagoya University Hospital, Ichinomiya Municipal Hospital, Japanese Red Cross Nagoya First Hospital, and Komaki City Hospital. The present study was

approved by the Institutional Review Board of each participating institution and all patients provided informed consent for banking and molecular analyses in accordance with the Declaration of Helsinki. Mononuclear cells isolated from the fresh BM or PB samples of primary acute leukemia patients were intravenously injected into 6-week-old NOG mice (purchased from the Central Institute for Experimental Animals, Tokyo, Japan) at $1–15 \times 10^6$ viable cells per mouse. T cells from patient BM samples were depleted by intraperitoneally injecting an anti-human CD3 (OKT3) antibody (Exbio Antibodies, Prague, Czech Republic). NOG mice were not pre-conditioned with irradiation. NOG mice were maintained in a 12-hour dark/light cycle, at a temperature of $23 \pm 2\ °C$ and a humidity of $55 \pm 10\%$. All animal procedures were approved by the Institutional Animal Care and Use Committee of Nagoya University (M210583-002) and carried out in accordance with the Regulations on Animal Experiments in Nagoya University.

**Engraftment analysis of PDX models**. Primary AML cells were transplanted into one NOG mouse in each case, except for five cases in which AML cells were transplanted into duplicated mice (Supplementary Fig. S3). The engraftment of primary acute leukemia cells was monitored every 3 weeks in PB from the tail vein followed by flow cytometry analyses using FACSAria2 (BD Biosciences, San Jose, CA, USA) with anti-mouse CD45-PerCP (30-F11) (BioLegend, San Diego, CA, USA), anti-human CD3-APC (UCHT1), and anti-human CD45-PE (HI30) antibodies (BD Biosciences, San Jose, CA, USA). Mice were sacrificed when PB human CD45$^+$ reached >0.5% at 2 time points, followed by a flow cytometry assessment of the PB, spleen, and BM for engrafted human cells with the same antibodies. The engraftment of the primary sample was confirmed when the BM human CD45$^+$ percentage was higher than 20% and serial transplantation at $1 \times 10^6$ cells was successful, whereas engraftment failure was confirmed when PB human CD45$^+$ did not increase to 0.5% until 365 days post-transplantation of primary leukemia cells or the serial transplantation of $1 \times 10^6$ PDX BM cells, or when the human CD45$^+$ percentage in harvested BM was lower than 20%. In AML, the presence of human CD45$^+$CD3$^+$ cells in harvested BM was also regarded as the engraftment failure of primary leukemia cells. After the confirmation of engraftment by successful serial transplantation, primagraft or secondary passaged (P1-2) BM samples were used in the mutational or flow cytometric analysis.

**Flow cytometry analysis**. Cryopreserved primary AML cells and engrafted PDX BM cells were thawed and their phenotypes were analyzed using FACSAria2 (BD Biosciences, San Jose, CA, USA) with anti-mouse CD45-APC/Cy7 (30-F11, dilution 1:100), anti-human CD45-PerCP/Cy5.5 (HI30, dilution 1:100) (BioLegend, San Diego, CA, USA), anti-human CD45-PE (HI30, dilution 1:100), anti-human CD34-APC (8G12, dilution 1:100), anti-human CD38-PE/Cy7 (HB7, dilution 1:100), and anti-human CD3-APC (UCHT1, dilution 1:100) antibodies (BD Biosciences, San Jose, CA, USA). Flow cytometry data were collected using FACSAria2 and FACSDiva software v8.0.1 (BD Biosciences, San Jose, CA, USA) and analyzed using FlowJo software v8.8.7 (BD Biosciences, San Jose, CA, USA). The gating strategy is detailed in Supplementary Fig. 6.

**Mutation analysis**. Pairs of transplanted primary BM or PB samples and BM samples of their PDX were subjected to a sequencing analysis. Genomic DNA was extracted from unfractionated primary BM or PB samples and PDX BM samples using the QIAamp DNA Blood Mini Kit or QIAamp DNA investigator Kit (QIAGEN, Hilden, Germany). The human CD45$^+$ fraction was sorted from PDX BM when it was less than 90% by magnetic cell separation using MACS MicroBeads (human CD45 MicroBeads; Miltenyi Biotec, Bergisch Gladbach, Germany) following the manufacturer's recommendations.

The target sequencing of 54 genes, which are frequently identified in the presence of myeloid malignancies, was performed using the TruSight Myeloid Sequencing Panel and Illumina MiSeq sequencer according to the manufacturer's instructions (Illumina, San Diego, CA)[38]. Variant Studio software version 3.0 (Illumina, San Diego, CA) was used and targeting regions with a sequencing depth of more than 100 were considered to be suitable for analysis. Detected genomic lesions that had either been previously published as recurrent in leukemia from the literature or the public database COSMIC or were predicted to alter protein function on tumor suppressor genes were designated as significant somatic mutations. The internal tandem duplication of the FLT3 gene (FLT3-ITD) was examined by a fragment analysis as previously reported[39]. The functional domains of FLT3 gene were PCR-amplified with forward primers 5′ end labeled with fluorescent dye using AmpliTaq Gold DNA Polymerase with Gold Buffer and $MgCl_2$ (Applied Biosystems, Foster City, CA) according to the manufacturer's instructions. The PCR products were analyzed using ABI 3500 Genetic Analyzer and 3500 Series Data Collection Software version 1.0 (Applied Biosystems, Foster City, CA) and the amplicons with a size greater than that of wild type were interpreted as positive for the ITD mutation. The number, area and length of mutant peaks on the electropherogram were analyzed using GeneMapper analysis software version 4.1 (Applied Biosystems, Foster City, CA). The primer sequences can be found in Supplementary Table 6.

Germline samples were absent and, thus, not analyzed.

**Reverse transcription PCR.** RNA was extracted from unfractionated primary BM or PB samples and PDX BM samples using the QIAamp RNA Blood Mini Kit (QIAGEN, Hilden, Germany). Complementary DNA was generated from total RNA using SuperScript II Reverse Transcriptase (Invitrogen, Carlsbad, CA) according to the manufacturer's instructions. We examined chimeric gene transcripts (Major *BCR-ABL1*, Minor *BCR-ABL1*, *PML-RARA*, *CBFB-MYH11*, *MLL-AF4*, *MLL-AF9*, *MLL-ENL*, and *MLL-ELL*) by reverse transcription PCR using AmpliTaq Gold DNA Polymerase with Gold Buffer and MgCl₂ (Applied Biosystems, Foster City, CA) according to the manufacturer's instructions. Amplification was evaluated by agarose gel electrophoresis[40]. The primer sequences can be found in Supplementary Table 6.

**Statistical analysis.** Demographic factors and disease characteristics were compared using the standard $\chi^2$ test or Fisher's exact test for categorical data and the Mann–Whitney $U$-test for continuous variables. The relationship between patient and PDX VAF was assessed using Pearson's correlation coefficient and $P$-values. Changes in VAF between patients and PDX were examined by subtracting the VAF of patients from that of PDX. The probabilities of EFS were estimated using the Kaplan–Meier method, and the Log-rank test was used for univariate comparisons. Survival times were calculated from the day of achieving CR to relapse, death due to any cause, or the last follow-up. $P < 0.05$ was considered to be significant for all analyses. All statistical analyses were performed by Stata version 12 (Stata Corp., College Station, TX) and GraphPad Prism 6 (GraphPad Software Inc., San Diego, CA).

**Reporting summary.** Further information on research design is available in the Nature Research Reporting Summary linked to this article.

## Data availability

The COSMIC database used to define significant somatic mutations can be found at https://cancer.sanger.ac.uk/cosmic. The datasets of targeted sequencing generated and analyzed during the current study are available under restricted access in compliance with patient consent for data sharing, access can be obtained by approval from the Institutional Review Board of Nagoya University and each participating institution, namely Ichinomiya Municipal Hospital, Japanese Red Cross Nagoya First Hospital and Komaki City Hospital; (Contact person: Hitoshi Kiyoi, Email: kiyoi@med.nagoya-u.ac.jp). There are no restrictions on who will be granted access to this data, and data will be provided within a month of request. Source data are provided with this paper.

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

## Acknowledgements

We would like to acknowledge Yohei Yamaguchi, who sadly passed away, for his significant contributions to this research work. The present study was supported by Grants-in-Aid from the Practical Research for Innovative Cancer Control from the Japan Agency for Medical Research and Development, AMED (17ck0106251h0001, 18ck0106251h0002, 19ck0106251h0003, 20ck0106535h0001, and 21ck0106535h0002), the Project for Development of Innovative Research on Cancer Therapeutics (P-DIRECT) from AMED (19cm0106562h0001, 20cm0106562h0002, and 21cm0106581h0001), and the Scientific Research Program of the Ministry of Education, Culture, Sports, Science and Technology of Japan (17K09921).

## Author contributions

N.K., Y.I., and H.K. designed the study and interpreted the data; N.K., Y.I., and H.K. wrote the manuscript; N.K., A.A., J.K., Y.Y., H.H., Y.U., and Y.I. performed molecular analyses; N.K., Y.I., Y.U., M.N., S.I., R.K., T.N., T.M., K.W., Y.O., and K.K. collected samples and clinical data and critically reviewed the draft; and all authors approved the final version submitted for publication, with the exception of Y.Y. (deceased during the peer review).

## Competing interests

H. Kiyoi received research funding from FUJIFILM Corporation, Kyowa Hakko Kirin Co., Ltd., Bristol-Myers Squibb, Otsuka Pharmaceutical Co., Ltd., Perseus Proteomics Inc., Daiichi Sankyo Co., Ltd., Abbvie Inc., CURED Inc., Astellas Pharma Inc., Chugai Pharmaceutical Co., Ltd., Zenyaku Kogyo Co., Ltd., Nippon Shinyaku Co., Ltd., Eisai Co., Ltd., Takeda Pharmaceutical Co., Ltd., Sumitomo Dainippon Pharma Co., Ltd., Novartis Pharma K.K., Sanofi K.K., and Pfizer Japan Inc., and honoraria from Bristol-Myers Squibb, Astellas Pharma Inc., and Novartis Pharma K.K. The remaining authors declare no competing interests. All other authors have no competing non-financial interests to declare.
