## [Peer Review File · Nature Communications]

Comparison of clonal architecture between primary and immunodeficient mouse-engrafted acute myeloid leukemia cellsREVIEWER COMMENTS

Reviewer #1 (Remarks to the Author): Expert in AML PDX models

Kawashima et al. performed an impressive number of xenografts of patients (PDX) with acute myeloid leukemia (AML) in immune incompetent NOG mice and determined the clonal architecture by sequencing for somatic mutations. They found that engraftment was linked to whether the samples were de novo or relapse. In addition, some mutations (FLT3, NPM1, and others) were more prevalent in samples that successfully engrafted (Fig.1). The status of CD34/CD38 surface expression was not concordant in the majority of the cases. In addition, the variant allele frequency (VAF) of particular genetic variants (e.g. FLT3-ITD/TKD, N/K-RAS, DNMT3A) was also not consistent in cells before and after engraftment supporting clonal selection (Fig.2). Comparison of mutation VAF of de novo diagnosed vs. relapse/refractory AML before and after engrafting revealed that the latter show more concordant clone size (Fig.3). Serial propagation of PDX into secondary recipients revealed differential increase, decrease or stability in samples carrying specific mutations (Fig.4). PDX-engraftment was associated with a lower probability of event-free survival (Fig.5). Finally, they evaluated clonal composition in AML samples at diagnosis and relapse with the one observed after PDX. Interestingly, the patients relapse clones were often selected during the primary PDX (Fig.6). These data let them to conclude "that the PDX-specific enrichment of AML clones has predictive power for characterizing clonal selection" Overall it is an interesting and easily accessible paper. Clarification of some emerging questions would further increase the quality of this manuscript.

Major points

1. Similar work has recently been published in Nature Communications (<https://doi.org/10.1038/s41467-019-14106-0>). The authors should compare their data with the data from Sanden et al. These researchers used NSG mice raising the question whether such data can be directly be compared? Could the authors may be repeat a few PDX in NSG and compare the clonal composition?
2. Serial mutational analysis of AML cells from patients at diagnosis and relapse followed by PDX revealed consistence between dominant clones in the first PDX and the clone than ultimately expanded upon relapse in patients. The authors claim to have analyzed 21 patients but they show only data from 5 patients? The reviewer wonders whether the data would sufficient to develop a predictive model that could be validated in a blinded fashion?

Minor points

1. Line 100: "No significant differences were observed in primary diseases, graft sources or infused blast percentages". The primary diseases seem rather different (based on FAB)? What do they mean with that? Patient blood counts, health status or what?
2. Line 103: "median 5.0 vs. 5.0 x 10⁶, p=0.011" is confusing: can one not show this differently? e.g. the rate of success vs. failure below 5.0 and above 5.0 (resulting in a given p-value)?
3. There was a wide variation of engraftment latency (27-380days): any correlations to clinico-pathological characteristics of the patient's disease?
4. Line 109: "among 109 transplanted AML cells"...is most likely not correct and should rather be "cells from 109 AML patients".
5. Figure 3a/b: a significant difference between a (primary diagnosed) and b (refractory/relapse) samples is not obvious in these graphs?
6. Fig. 4: a) maybe one can use 3 different colors for those samples that increased, decreased or remained stable over serial passaging. In contrast, the colors in 4b-d) appear rather confusing: maybe use again the three colors and list the gene with the % given for each value?
7. Serial propagation of AML PDX revealed a disconcordant behavior of K- and N-RAS mutations: any explanation?
8. Fig.5 shows the association of event-free survival with positive or negative engraftment. What about overall survival, chemoresistance or time to relapse? Was there also a correlation with the numbers of subclones (>5%VAF) detected before PDX?
9. Fig.6 shows clonal dynamics between patient diagnosis and relapse and PDX from diagnosis and relapse. The panel showing different VAF (%) starts with diagnosis PDX...followed by diagnosis Pat...but should this not be the other-way-round? The disease develops in the patient BEFORE being modeled in the mouse?

Reviewer #2 (Remarks to the Author): Expert in clonal heterogeneity and leukaemia genomics

Kawashima and colleagues report on an impressive effort to generate and genomically characterize a large cohort of AML PDXs. Initially they attempted to engraft 160 AMLs and they were successful (as defined by >20% hCD45 in the BM and the ability to serial passage) in 105. Overall this is an impressive study considering its sheer number of different PDXs. However, many of the conclusions have already been described in other, albeit smaller, studies. Overall there are concerns about novelty. In addition, many of their conclusions lack statistical support.

Specific Comments:

1. The statement in the second sentence of the abstract is not accurate. This has been addressed by many other studies, which does question the novelty of this work.
2. In the abstract, the authors state that "PDX models may make the clonal hierarchy of heterogeneous AML more prominent". I do not understand this statement. This is repeated on line 358.
3. Line 58. What do the authors mean that "clonal diversity occurs at the time of recurrence"?
4. Regarding the experiment design.
 - a. How many mice were attempted for each AML? If more than 1, was an AML deemed engraftable if just one mouse or all the mice fulfilled the criteria for engraftment?
 - b. Please be clear regarding the sequencing of samples. Engraftability required serial passaging, but for most of the genomic studies I am under the impression that the primagraft was used. Please confirm.
 - c. Were the recipient NOG mice pre-conditioned with radiation?
 - d. A breakdown of the failures is needed. How many didn't engraft at all in the primary at 365 days, how many failed serial passaging, how many had T cell engraftment and how many had >5% in the blood but <20% in the bone marrow?
5. Lines 99-100. Please show these data with p values.
6. The number of samples is somewhat confusing. Is there a reason to include the 45 samples with no genomic data from patients at all in this study?
7. Lines 143-150. The use of initiating versus driver mutations is confusing. Plus, most would consider DNMT3A to be an initiating event in this nomenclature.
8. There is no mention of fusion events in this study. Clearly some of these have recurrent fusions such as CBF-AMLs or KMT2Ar?
9. One of the main arguments is that there are fewer clones at relapse. However, a statistically test is not used to prove this argument in Figure 3. Further in line 166 they say that genetic background with primary samples and PDX is more concordant at relapse than diagnosis also without a statistical test.
10. Another main argument is that patients with AMLs that engraft have a worse outcome (Figure 5). Did the authors look at other known genetic or clinical factors associated with AML outcome? Isn't it possible that the failure of low risk subtypes (CBF-AML) to engraft had nothing to do with clonal heterogeneity? Or that patients with more aggressive disease (because of high-risk lesions) were more likely to engraft?
11. Lines 234-236. The authors claim that 12 of 21 are consistent and they reference Figure 6. Figure 6 shows presentative cases but not the data needed to make the 12 of 21 statement.
12. Some of the statements are over generalizing the data. Ultimately there are few rules with PDXs and each sample is unique. Specifically see line 287. In addition, line 126 since the authors only performed a limited analysis of the PDXs.
13. Figure 4a. What do the authors mean by "each dot plot"? Also, I am not certain of the number in the middle of the pie chart and the overall significance of the pie charts in panels B-D.
14. Line 88, the authors never address the "proliferative potential" of AML cells.
15. Line 101, the authors say, "Engrafted primary samples were derived from higher numbers of patients with AML," but there is no p value shown in table 1.
16. Please increase the font size for the text in Figure 6.
17. Line 117, the authors describe that some genes accumulated more in patients with engraftment. However, many of these (ASXL1 for example) show no significant difference between with or without engraftment.

Reviewer #3 (Remarks to the Author): Expert in AML genomics

In this study, Kawashima and colleagues used 160 adult AML patient samples to generate patient derived xenotransplant (PDX) models in NOD/Shi-scid, IL-2R γ null (NOG) mice. Of these cases, 105 AMLs engrafted successfully, and by tracking the dynamics of somatic mutations in serial samples from primary AML cells and the respective PDX models, the authors could provide further evidence for a PDX-specific enrichment of AML clones. This enrichment has predictive power as it is characteristic of the clonal selection in parental patients.

Major comments:

- Preclinical mouse models are of great value for an improved understanding of AML. While many groups are working with respective models, there also have been many very good publications addressing the role and value of mouse models, including PDX models (see e.g. Almosailekh and Schwaller, *Int J Mol Sci* 2019). In my opinion, the introduction falls a little short in mentioning respective models and previous efforts.
- Similarly, important work was not cited/discussed in the work by Kawashima and colleagues. For example, the work by Culen and colleagues on the influence of mutational status and biological characteristics of acute myeloid leukemia on xenotransplantation outcomes in NOD SCID gamma mice (*J Cancer Res Clin Oncol* 2018). Or the work by Paczulla et al. showing that long-term observation reveals high-frequency engraftment of human acute myeloid leukemia in immunodeficient mice (*Haematologica* 2017), and the study by Griessinger and colleagues on acute myeloid leukemia xenograft success prediction (*Exp Hematol* 2018).
- Where there differences with regard to ELN risk groups for the engraftment? In accordance, ELN risk classification and cytogenetic information should be included in Table 1.
- The authors do nicely demonstrate that more aggressive leukemias with ELN high risk markers such as FLT3, ASXL1, and RUNX1 are enriched among the engrafted cases. What about TP53? If it is not enriched, what could be the reason, what do the authors think about this?
- There was a marked discordance with regard to the immunophenotype CD34/CD38 among cases. With that regard it would be good to refer to the work by John Dick's group who did link the respective quadrants with "leukemia stemness", some discussion with that regard would be interesting to the readers (see Ng et al. *Nature* 2016).
- Finally, with regard to the clonal selection in PDX models, the VAF analyses provided by the authors are quite interesting, although some interpretations need further clarification/revision. For example, on p.7 the authors state that "VAF was unstable between primary AML and PDX cells in gene mutations considered to be driver mutations", i.e. rather late events. However, the authors do list CEBPA which is rather a transforming early event, and they do also list DNMT3A which is a very early mutation predisposing to AML and which can also be found in healthy individuals. How can it be that these mutations are also enriched? Furthermore, the VAF for CEBPA did also diminish in serial transplantation, correct?
- And with regard to the correlation with clinical parameters, I was wondering why the clinical information was not provided for all de novo AMLs, did the cases not mentioned not receive intensive chemotherapy? And was there an impact of age on the PDX outcome?

Minor comments:

- p.5: "... at relapse/refractory ($P < 0.001$) with higher numbers of infused cell counts (median, 5.0×10^6 vs 5.0×10^6 cells, $P = 0.011$) than those in failure." Please double check whether this statement is correct, 5 vs 5 is not higher, right?

Response to referees

We appreciate the careful review by the Editors and the reviewers of our manuscript.

First of all, we apologize for the delay in resubmission. According to the suggestion of the reviewer, we generated additional PDX models using NSG mice to compare clonal compositions of our NOG models to previous studies. Due to the Covid-19 pandemic, availability of experiment materials, especially sources for animal experiments were very limited in our institution. Therefore, completing additional experiments took long period of time.

We have revised our manuscript according to the recommendations of the reviewers. Here we include a point-by-point response to each of the reviewers' comments and provide a revised version of our manuscript with highlighted changes.

We think that integrating these points has improved the manuscript and hope that the manuscript is now suitable for publication in *Nature Communications*.

REVIEWER COMMENTS

Reviewer #1 (Remarks to the Author): Expert in AML PDX models

Kawashima et al. performed an impressive number of xenografts of patients (PDX) with acute myeloid leukemia (AML) in immune incompetent NOG mice and determined the clonal architecture by sequencing for somatic mutations. They found that engraftment was linked to whether the samples were de novo or relapse. In addition, some mutations (FLT3, NPM1, and others) were more prevalent in samples that successfully engrafted (Fig.1). The status of CD34/CD38 surface expression was not concordant in the majority of the cases. In addition, the variant allele frequency (VAF) of particular genetic variants (e.g. FLT3-ITD/TKD, N/K-RAS, DNMT3A) was also not consistent in cells before and after engraftment supporting clonal selection (Fig.2). Comparison of mutation VAF of de novo diagnosed vs. relapse/refractory AML before and after engrafting revealed that the latter show more concordant clone size (Fig.3). Serial propagation of PDX into secondary recipients revealed differential increase, decrease or stability in samples carrying specific mutations (Fig.4). PDX-engraftment was associated with a lower probability of event-free survival (Fig.5). Finally, they evaluated clonal composition in AML samples at diagnosis and relapse with the one observed after PDX. Interestingly, the patients relapse clones were often selected during the primary PDX (Fig.6). These data let them to conclude “that the PDX-specific enrichment of AML clones has predictive power for characterizing clonal selection” Overall it is an interesting and easily accessible paper. Clarification of some emerging questions would further increase the quality of this manuscript.

Major points

1. Similar work has recently been published in Nature Communications (<https://doi.org/10.1038/s41467-019-14106-0>). The authors should compare their data with the data from Sanden et al. These researchers used NSG mice raising the question whether such data can be directly be compared? Could the authors may be repeat a few PDX in NSG and compare the clonal composition?

According to the reviewer's suggestion, we transplanted frozen and fresh primary AML cells from 6 and 3 patients into 9 NSG mice, respectively. Engraftment of human CD45+ cells was observed in 1 of 9 NSG mice. The following results of mutational analysis revealed that the VAF changes between the primary patient to NOG was consistent with those to NSG. Therefore, the data of studies using NSG mice could be directly compared to ours, however the number of the comparable PDX models was limited.

Gene	Amino Acids	CDS Position	Variant	Codons	Patient VAF (%)	P1NOG VAF (%)	P1NSG VAF (%)	Consequence
KRAS	G12V	35	C>C/A	gGt/gTt	38.28	37.21	42.48	missense variant
GATA2	T354T CQT	1061-10 62	C>C/C GTCTG ACAA	acg/acTTGT CAGACg	48.35	55.71	48.39	inframe insertion
KRAS	G12D	35	C>C/T	gGt/gAt	-	7.83	6.16	missense variant

2. Serial mutational analysis of AML cells from patients at diagnosis and relapse followed by PDX revealed consistence between dominant clones in the first PDX and the clone than ultimately expanded upon relapse in patients. The authors claim to have analyzed 21 patients but they show only data from 5 patients? The reviewer wonders whether the data would sufficient to develop a predictive model that could be validated in a blinded fashion?

We thank the reviewer for this suggestion. We added the data of clonal changes in the rest of 16 patients as Supplementary Figure 5.

Minor points

1. Line 100: "No significant differences were observed in primary diseases, graft sources or

infused blast percentages”. The primary diseases seem rather different (based on FAB)? What do they mean with that? Patient blood counts, health status or what?

We intended to mention there was no difference in the number of patients according to de novo AML or not, however this was very confusing as the reviewer kindly pointed out. We rephrased this sentence to “Successful engraftment was also associated with the M4-FAB type (P=0.028), relapse/refractory (P<0.011) and higher risk of the European LeukemiaNet (ELN) classification (adverse, 74.7%; intermediate, 55.2%; favorable, 41.7%; P=0.022), but not with patients’ age (P=0.621), cytogenetic risks (P=0.053), graft sources (P=0.060), and infused blast percentages (P=0.280).” (lines 113-117, page 5).

2. Line 103: “median 5.0 vs. 5.0×10^6 , p=0.011” is confusing: can one not show this differently? e.g. the rate of success vs. failure below 5.0 and above 5.0 (resulting in a given p-value)?

We thank the reviewer for this suggestion. We revised this sentence as follow; “Engraftment was achieved within a median of 106 days (range, 27-380 days) post-transplantation, and was more frequently observed in more than 5.0×10^6 cells transplanted mice (71% vs. 38%, P=0.001).” (lines 111-113, page 5)

3. There was a wide variation of engraftment latency (27-380days): any correlations to clinico-pathological characteristics of the patient’s disease?

We summarized the correlation between patients’ clinicopathological characteristics and the engraftment latency of their PDX models in Supplementary Table 4 and the following description (lines 196-201, page 9).

“Engraftment latency was widely ranged from 27 to 380 days. Time to engraftment was associated with the ELN risk categories (adverse risk group; median 104 days, range 45-362 days, intermediate risk group; median 126 days, range 27-196 days, favorable risk group; median 161 days, range 106-330 days, P=0.036). On the other hand, it was not associated with FAB classifications, cytogenetic abnormalities and infused blast burden (Supplementary Table 4).”

4. Line 109: “among 109 transplanted AML cells”...is most likely not correct and should rather be “cells from 109 AML patients”.

We revised this sentence to “Among cells from 160 AML patients, ...” (line 122, page 6).

5. Figure 3a/b: a significant difference between a (primary diagnosed) and b (refractory/relapse) samples is not obvious in these graphs?

We revised the description of this result and also added Figure 3c for better understanding. As mentioned in lines 177-181, page 8, “Although changes in the VAF were observed in many PDX models both of newly diagnosed and R/R samples, a 10-fold or higher increase in the VAF was more frequently observed in PDX models of newly diagnosed samples (22/169 variants) than those of R/R samples (8/137 variants, $P=0.033$, Figure 3c).”

6. Fig. 4: a) maybe one can use 3 different colors for those samples that increased, decreased or remained stable over serial passaging. In contrast, the colors in 4b-d) appear rather confusing: maybe use again the three colors and list the gene with the % given for each value?

We thank the reviewer for this suggestion. We revised Figure 4 for the better understanding of 3 types of clonal changes as the reviewer suggested.

7. Serial propagation of AML PDX revealed a discordant behavior of K- and N-RAS mutations: any explanation?

We added the following explanation in Discussion session (lines 329-339, page14); “In contrast to the NRAS mutated clones, the KRAS mutated clones expanded during serial propagation of AML-PDX (Figure 4b, d). Since all KRAS mutated and expanded clones were thought to be co-mutated with WT1 according to their VAFs, these clones might expand through propagations. In the NRAS mutated primary samples, FLT3-ITD, TET2, TP53 or WT1 mutation was also identified; however, VAFs of NRAS were different from those of the other genes indicating that all NRAS mutated clones did not harbor FLT3-ITD, TET2, TP53 or WT1 mutation. In 16 NRAS engrafted P1-2 PDX models, 11 showed stable or enriched engraftment of the NRAS mutated clones, whereas NRAS mutated clones were drastically diminished in the other 5 models because the clones carrying FLT3-ITD (N=4), TET2 (N=2), TP53 (N=1) or WT1 (N=1) dominantly expanded.”

8. Fig.5 shows the association of event-free survival with positive or negative engraftment. What about overall survival, chemoresistance or time to relapse? Was there also a correlation with the numbers of subclones (>5%VAF) detected before PDX?

There was no association between the numbers of subclones in either patients or PDX and EFS, OS, or relapse rate. We added the following data in the Result section (lines 239-243, page 10); “Overall survival (OS) rate and cumulative incidences of relapse were also significantly poorer in PDX-engrafted patients, which were 28.8% (95%CI, 10.5-50.3%) vs. 50.9% (95%CI, 22.3-73.9%, $P=0.045$), and 74.6% (95%CI, 54.8-86.7%) vs 45.4% (95%CI, 27.6-61.6%, $P=0.029$) at 2 years, respectively (Supplementary Figure 4).”

9. Fig.6 shows clonal dynamics between patient diagnosis and relapse and PDX from diagnosis

and relapse. The panel showing different VAF (%) starts with diagnosis PDX...followed by diagnosis Pat...but should this not be the other-way-round? The disease develops in the patient BEFORE being modeled in the mouse?

We revised Figure 6 as the reviewer suggested.

Reviewer #2 (Remarks to the Author): Expert in clonal heterogeneity and leukaemia genomics

Kawashima and colleagues report on an impressive effort to generate and genomically characterize a large cohort of AML PDXs. Initially they attempted to engraft 160 AMLs and they were successful (as defined by >20% hCD45 in the BM and the ability to serial passage) in 105. Overall this is an impressive study considering its sheer number of different PDXs. However, many of the conclusions have already been described in other, albeit smaller, studies. Overall there are concerns about novelty. In addition, many of their conclusions lack statistical support.

Specific Comments:

1. The statement in the second sentence of the abstract is not accurate. This has been addressed by many other studies, which does question the novelty of this work.

We revised this description and further discussed the novelty of this work compared to previous studies in Discussion (lines 27-30 on page 2, lines 340-356 on pages 14-15, lines 367-374 on pages 15-16).

2. In the abstract, the authors state that “PDX models may make the clonal hierarchy of heterogeneous AML more prominent”. I do not understand this statement. This is repeated on line 358.

We apologize for this misleading description.

We rephrased this sentence to “primary AML cells including potentially chemotherapy-resistant clones dominantly engraft in AML-PDX models” (lines 36-37 on page 2 in Abstract and lines 412-413 on page 17 in Discussion).

3. Line 58. What do the authors mean that “clonal diversity occurs at the time of recurrence”?

We revised this description as follows; “Although multiple subclones proliferate in a patient, relapse is based on a selection of existing clones or an acquisition of additional mutations during the progression of the disease” (lines 65-67, page 3).

4. Regarding the experiment design.

a. How many mice were attempted for each AML? If more than 1, was an AML deemed engraftable if just one mouse or all the mice fulfilled the criteria for engraftment?

Primary AML cells were transplanted into one NOG mouse in each case except for 5 cases that AML cells were transplanted and successfully engrafted in duplicated mice in every case (Supplementary Figure S3). We additionally described this in Methods (lines 435-436, page 18)

b. Please be clear regarding the sequencing of samples. Engraftability required serial passaging, but for most of the genomic studies I am under the impression that the primagraft was used. Please confirm.

We apologize for unclear description. We added this explanation in Methods (lines 450-452, page 19); “After the confirmation of engraftment by their successful serial transplantation, their primagraft or secondary passaged (P1-2) BM samples were used for mutational or flow cytometric analysis.”

c. Were the recipient NOG mice pre-conditioned with radiation?

NOG mice were not pre-conditioned with irradiation. We described this in Methods (lines 430, page 18)

d. A breakdown of the failures is needed. How many didn't engraft at all in the primary at 365 days, how many failed serial passaging, how many had T cell engraftment and how many had >5% in the blood but <20% in the bone marrow?

We thank the reviewer for this suggestion. We described the breakdown of engraftment failure in Result section “28 models died within 365 days (median 270 days, range 110-364) without positive hCD45+ in PB, 18 with negative human CD45+ in PB until day 365, 7 with T cell engraftment, and 2 failed in serial passaging with hCD45+>5% in the blood but <20% in the bone marrow at day 365.” (lines 107-110, page 5)

5. Lines 99-100. Please show these data with p values.

We added P-values in Table 1 and lines 114-118 on page 5; “Successful engraftment was also associated with the M4-FAB type (P=0.028), relapse/refractory (P<0.001) and higher risk of the European LeukemiaNet (ELN) classification (adverse, 74.7%; intermediate, 55.2%; favorable, 41.7%; P=0.022), but not with patients' age (P=0.621), cytogenetic risks (P=0.053), graft sources (P=0.060), and infused blast percentages (P=0.280).”

6. The number of samples is somewhat confusing. Is there a reason to include the 45 samples with no genomic data from patients at all in this study?

We apologize for confusing description. A total of 160 patients whose clinical characteristics were available were included in the analysis of clinical features of AML cells associated with engraftment into PDX (Table 1).

We described the reasons for excluding 45 patients with no genomic data more clearly as follows: “The other 36 AML samples with engraftment and 9 samples with engraftment failure were excluded from the following genetic analysis because the quality of patients’ DNA was insufficient or they were derived from different time points of the same patients” (in lines pages 125-128 on page 6).

7. Lines 143-150. The use of initiating versus driver mutations is confusing. Plus, most would consider DNMT3A to be an initiating event in this nomenclature.

We apologize for confusing description. We deleted this description from result section on pages 7-8.

8. There is no mention of fusion events in this study. Clearly some of these have recurrent fusions such as CBF-AMLs or KMT2Ar?

We thank the reviewer for this comment. Chimeric transcripts were detected using RT-PCR as mentioned in Methods (lines 477-484, page 20). We added impact of fusion events on PDX engraftment and changes between patients and PDX AML cells in Results section as follows: “AML cells with RUNX1-RUNX1T1 (0% vs. 7%, $P=0.03$) and CBFβ-MYH11 (1% vs. 9%, $P=0.06$) showed lower engraftment rates” (lines 134-135, pages 6) and “In 11 patients with chimeric transcripts (Figure 1), they were preserved in all engrafted cells.”. (line 170, page 8).

9. One of the main arguments is that there are fewer clones at relapse. However, a statistically test is not used to prove this argument in Figure 3. Further in line 166 they say that genetic background with primary samples and PDX is more concordant at relapse than diagnosis also without a statistical test.

We apologize for misleading description. We revised the description of this result and also added Figure 3c for better understanding. As we additionally described in lines 177-182 on page 8, “Although changes in the VAF were observed in many PDX models both of newly diagnosed and R/R samples, a 10-fold or higher increase in the VAF was more frequently observed in PDX models of newly diagnosed samples (22/169 variants) than those of R/R samples (8/137 variants, $P=0.033$, Figure 3c). These results

suggested that newly diagnosed AML cells contained more minor clones which have a growth potential in NOG mice than R/R samples.”

10. Another main argument is that patients with AMLs that engraft have a worse outcome (Figure 5). Did the authors look at other known genetic or clinical factors associated with AML outcome? Isn't it possible that the failure of low risk subtypes (CBF-AML) to engraft had nothing to do with clonal heterogeneity? Or that patients with more aggressive disease (because of high-risk lesions) were more likely to engraft?

We thank the reviewer for this suggestion. We analyzed the impact of clinical characteristics on AML outcome in addition to PDX engraftment status. Multivariate analysis revealed that patients' older age and ELN adverse risk subgroup as well as PDX engraftment were risk factors for patients' survival, however PDX engraftment showed the highest hazard ratio.

Variables		Univariate			Multivariate		
		HR	(95%CI)	P-value	HR	(95%CI)	P-value
Patient age	>=60 years	3.39	(1.72-6.67)	<0.001	2.93	(1.46-5.87)	0.003
Disease	M3	0.65	(0.16-2.70)	0.551			
Karyotype	Poor	1.29	(0.64-2.61)	0.472			
ELN risk category*	Adverse	2.28	(1.20-4.32)	0.012	1.98	(1.01-3.86)	0.046
Graft source	PB	1.54	(0.74-3.19)	0.245			
Infused cell count	>= 5 x10 ⁶	1.29	(0.54-3.10)	0.566			
PDX Engraftment	Engraftment	3.24	(1.64-6.41)	0.001	3.44	(1.73-6.85)	<0.001

*APL excluded

11. Lines 234-236. The authors claim that 12 of 21 are consistent and they reference Figure 6. Figure 6 shows presentative cases but not the data needed to make the 12 of 21 statement.

We thank the reviewer for this suggestion. We added the data of clonal changes in the rest of 16 patients as Supplementary Figure 5.

12. Some of the statements are over generalizing the data. Ultimately there are few rules with PDXs and each sample is unique. Specifically see line 287. In addition, line 126 since the authors only performed a limited analysis of the PDXs.

We revised these overgeneralizing expressions to the following descriptions.

Line 287 “In these cases, AML subclones selectively enriched in PDX models recapitulated clonal selection in patients acquiring treatment resistance.” (lines 310-312, page 13).

Line 126 “We then investigated whether AML-PDX recapitulated the molecular phenotypes of primary leukemia.” (lines 141-142, page 6)

13. Figure 4a. What do the authors mean by “each dot plot”? Also, I am not certain of the number in the middle of the pie chart and the overall significance of the pie charts in panels B-D.

We revised figure 4a to clarify dot plots indicating every variant. In addition, we showed them by different colors for better understanding of 3 types of clonal change as Reviewer#1 recommended. We also deleted the pie charts and showed the percentage of genes in figure 4b-d.

14. Line 88, the authors never address the “proliferative potential” of AML cells.

We apologize for inappropriate description and revised this sentence as follows; “We hypothesized that treatment-resistant clones in primary AML cells would engraft and expand in immunodeficient mice” (lines 96-97, page 5)

15. Line 101, the authors say, "Engrafted primary samples were derived from higher numbers of patients with AML," but there is no p value shown in table 1.

We showed each p-value in Table 1.

16. Please increase the font size for the text in Figure 6.

We increased the font size for the text in Figure 6.

17. Line 117, the authors describe that some genes accumulated more in patients with engraftment. However, many of these (ASXL1 for example) show no significant difference between with or without engraftment.

We revised this description as follows; “mutations in FLT3 (59% vs. 15%, $P<0.0001$), NPM1 (38% vs. 4%, $P<0.0001$), IDH1 (15% vs. 0%, $P=0.007$), and WT1 genes (13% vs. 2%, $P=0.04$) were more frequently observed in engrafted patients than in those with engraftment failure (Figure 1). Mutations in RAD21 (16% vs. 4%, $P=0.05$), ASXL1 (15% vs. 7%, $P=0.19$), RUNX1 (15% vs. 7%, $P=0.19$) and DNMT3A (25% vs. 15%, $P=0.22$) genes were also frequent, but there was no statistical significance.” (lines 129-134, page 6)

Reviewer #3 (Remarks to the Author): Expert in AML genomics

In this study, Kawashima and colleagues used 160 adult AML patient samples to generate patient derived xenotransplant (PDX) models in NOD/Shi-scid, IL-2R γ null (NOG) mice. Of these cases, 105

AMLs engrafted successfully, and by tracking the dynamics of somatic mutations in serial samples from primary AML cells and the respective PDX models, the authors could provide further evidence for a PDX-specific enrichment of AML clones. This enrichment has predictive power as it is characteristic of the clonal selection in parental patients.

Major comments:

- Preclinical mouse models are of great value for an improved understanding of AML. While many groups are working with respective models, there also have been many very good publications addressing the role and value of mouse models, including PDX models (see e.g. Almosaillekh and Schwaller, Int J Mol Sci 2019). In my opinion, the introduction falls a little short in mentioning respective models and previous efforts.

We thank the reviewer for this comment. We described and cited the previous efforts for AML mouse models in the introduction section as follows; “Since significant patient-to-patient cell heterogeneity complicates the clarification of a common mechanism that controls AML biology, a faithful model reflecting the complex heterogeneity of human AML in vivo would result in a better understanding of the molecular pathogenesis. Many groups have developed a variety of mouse leukemia models, such as those induced by chemicals, viral infection or irradiation, and transgenic animals expressing AML-associated proto-oncogenes. More recently, xenograft models using immuno-deficient mice enabled ex vivo maintenance and expansion of primary leukemia cells.” (lines 55-61, page 3)

- Similarly, important work was not cited/discussed in the work by Kawashima and colleagues. For example, the work by Culen and colleagues on the influence of mutational status and biological characteristics of acute myeloid leukemia on xenotransplantation outcomes in NOD SCID gamma mice (J Cancer Res Clin Oncol 2018). Or the work by Paczulla et al. showing that long-term observation reveals high-frequency engraftment of human acute myeloid leukemia in immunodeficient mice (Haematologica 2017), and the study by Griessinger and colleagues on acute myeloid leukemia xenograft success prediction (Exp Hematol 2018).

We further discussed the novelty of our results compared with these previous works and cited them as references #26, 27 and 29 (lines 340-356 and 367-372, pages 15-16).

- Where there differences with regard to ELN risk groups for the engraftment? In accordance, ELN risk classification and cytogenetic information should be included in Table 1.

We thank the reviewer for this suggestion, we summarized ELN risk classification and cytogenetic information in Table 1.

We could additionally identify that higher ELN risk group patients showed higher frequency of engraftment (adverse 74.7%, intermediate 55.2%, favorable 41.7%, $P=0.022$) and described this in Result section (lines 113-117, page 5).

- The authors do nicely demonstrate that more aggressive leukemias with ELN high risk markers such as FLT3, ASXL1, and RUNX1 are enriched among the engrafted cases. What about TP53? If it is not enriched, what could be the reason, what do the authors think about this?

The VAF of *TP53* was elevated in PDX in 7 of 9 patients, however they have been already saturated to be higher than 50%, therefore, the difference between patient and PDX was small but they were both in high frequencies.

- There was a marked discordance with regard to the immunophenotype CD34/CD38 among cases. With that regard it would be good to refer to the work by John Dick's group who did link the respective quadrants with "leukemia stemness", some discussion with that regard would be interesting to the readers (see Ng et al. Nature 2016).

We thank the reviewer for this suggestion. We discussed this interesting point in Discussion session as follows; "It was reported that a frequency of leukemic long-term culture-initiating cells, but not extracellular phenotype, was associated with the engraftment in NSG mice.²⁷ The majority of CD34+ and the minority of CD34- fractions reportedly contained leukemia stem cells (LSC).²⁸ In our study, CD34-/38+ fraction dominantly expanded in most PDX models, while CD34+/CD38- or CD34+/38+ fraction decreased in engrafted PDX (Supplementary Table 2). In addition, an adverse risk of the ELN classification was associated with shorter engraftment latency as previously reported (Supplementary Table 4). These results collectively indicated that the cell fractions which are more differentiated than LSCs also have advantages on proliferation in PDX models if they acquire genetic alterations associated with clinical prognosis." (lines 347-356, page 15)

- Finally, with regard to the clonal selection in PDX models, the VAF analyses provided by the authors are quite interesting, although some interpretations need further clarification/revision. For example, on p.7 the authors state that "VAF was unstable between primary AML and PDX cells in gene mutations considered to be driver mutations", i.e. rather late events. However, the authors do list CEBPA which is rather a transforming early event, and they do also list DNMT3A which is a very early mutation predisposing to AML and which can also be found in healthy individuals. How can it be that these mutations are also enriched? Furthermore, the VAF for CEBPA did also diminish in serial transplantation, correct?

We thank the reviewer for this comment. CEBPA and DNMT3A mutations showed weak correlation and were enriched more in PDX than in primary AML cells (Figure 2e, g and Supplementary Figure 1b). As the reviewer pointed out, classifying by driver or initiating mutations was inappropriate therefore we deleted this description and added discussion on the higher engraftment ability of co-mutation of NPM1 and DNMT3A genes in lines 340-346 on pages 15-16.

- And with regard to the correlation with clinical parameters, I was wondering why the clinical information was not provided for all de novo AMLs, did the cases not mentioned not receive intensive chemotherapy? And was there an impact of age on the PDX outcome?

We apologize for confusing description on this point.

To evaluate the responses of patients to chemotherapy according to the potential to engraft in NOG mice, clinical outcomes were analyzed in 76 of 106 patients newly diagnosed with AML who received cytotoxic chemotherapies and whose primary AML cells at diagnosis were transplanted into PDX. The rest of 30 patients did not receive intensive chemotherapy.

With regard to the engraftment status, primary patients' age was similar between engraftment and failure groups. (Table1, lines 116-117, page 5)

Minor comments:

- p.5: “ ... at relapse/refractory (P<0.001) with higher numbers of infused cell counts (median, 5.0 vs 5.0 x 10⁶ cells, P=0.011) than those in failure.” Please double check whether this statement is correct, 5 vs 5 is not higher, right?

We thank the reviewer for this comment. We revised this sentence as follows; “Engraftment was more frequently observed in more than 5.0 x 10⁶ cells transplanted mice (71% vs. 38%, P=0.001)” (lines 112-113, page 5)

REVIEWERS' COMMENTS

Reviewer #1 (Remarks to the Author):

The authors have nicely answered to all my points resulting in a clearly improved MS. This work could further improve by writing edition by a professional service.

Reviewer #2 (Remarks to the Author):

The authors have sufficiently addressed my comments and concerns. This is a very strong revision.

Reviewer #3 (Remarks to the Author):

The revised version of the manuscript has improved significantly and the authors have successfully addressed all the comments raised by the reviewers.

Response to Reviewers' Comments

We appreciate the careful review by the Editors and the reviewers of our manuscript. We hope that the manuscript is now suitable for publication in *Nature Communications*.

Reviewer #1 (Remarks to the Author):

The authors have nicely answered to all my points resulting in a clearly improved MS. This work could further improve by writing edition by a professional service.

This manuscript was edited by a professional language editing service. The certificate of English editing is also attached.

Reviewer #2 (Remarks to the Author):

The authors have sufficiently addressed my comments and concerns. This is a very strong revision.

Reviewer #3 (Remarks to the Author):

The revised version of the manuscript has improved significantly and the authors have successfully addressed all the comments raised by the reviewers.